# Lipid Hormones at the Intersection of Metabolic Imbalances and Endocrine Disorders

**DOI:** 10.3390/cimb47070565

**Published:** 2025-07-18

**Authors:** Maria-Zinaida Dobre, Bogdana Virgolici, Ruxandra Cioarcă-Nedelcu

**Affiliations:** Department of Biochemistry, Faculty of Medicine, Carol Davila University of Medicine and Pharmacy, 050474 Bucharest, Romania; maria.dobre@umfcd.ro (M.-Z.D.); ruxinedelcu@gmail.com (R.C.-N.)

**Keywords:** glucocorticoids, thyroid hormones, androgens, MASLD, PCOS, insulin resistance

## Abstract

Lipid hormone imbalances involving glucocorticoids, thyroid hormones (THs), and sex hormones have widespread metabolic consequences, contributing to the global increase in obesity and insulin resistance. This review examines the complex role of disrupted lipid hormone pathways in the development of metabolic disorders, particularly metabolic dysfunction-associated steatotic liver disease (MASLD). Endocrine disorders such as hypercortisolism, hypothyroidism, and polycystic ovary syndrome (PCOS) are closely linked to MASLD through shared metabolic pathways. Mechanisms include glucocorticoid-induced gluconeogenesis and lipolysis, impaired lipid clearance in hypothyroidism, and the hyperandrogenism-induced downregulation of hepatic low-density lipoprotein (LDL) receptors. PCOS-related factors—such as central obesity, adipocyte hypertrophy, low adiponectin levels, and genetic predisposition—further promote hepatic steatosis. Thyroid dysfunction may also impair the hepatic deiodination of T4, contributing to lipid accumulation and inflammation. Given the overlapping pathophysiology among endocrine, hepatic, and reproductive disorders, multidisciplinary collaboration is essential to optimize diagnosis, treatment, and long-term cardiometabolic outcomes.

## 1. Introduction

Imbalances in steroid hormones—particularly glucocorticoids (GCs) and sex hormones—affect a wide array of tissues and metabolic pathways, underscoring their systemic influence. In recent years, the increased use of corticosteroids, combined with unhealthy dietary patterns and sedentary lifestyles, has emerged as a significant driver of the global rise in obesity [1]. This review explores the diverse mechanisms through which disruptions in lipid hormone signaling contribute to, or worsen, insulin resistance, with a particular emphasis on the liver’s central role in metabolic regulation. Although the term non-alcoholic fatty liver disease (NAFLD) is still widely used, the newer terms, metabolic dysfunction-associated fatty liver disease (MAFLD) and metabolic dysfunction-associated steatotic liver disease (MASLD) offer a more clinically meaningful framework [2].

Unlike NAFLD, MAFLD is diagnosed using a clear, inclusive, and positive set of criteria that do not require the exclusion of other liver diseases. The diagnosis is based on evidence of hepatic steatosis, confirmed via imaging or histology, along with at least one of the following: overweight or obesity, type 2 *diabetes mellitus*, or metabolic dysfunction—such as dyslipidemia, elevated blood pressure, or insulin resistance [3].

MASLD offers improved global consensus and standardized classification as part of the broader Steatotic Liver Disease system (SLD). It uses clearer, medical language (“steatotic” instead of “fatty”) and includes subtypes like MetALD (metabolic dysfunction with moderate alcohol use liver disease), enhancing diagnostic precision. By aligning with common cardiometabolic traits, it improves disease recognition and communication across specialties. Overall, MASLD refines MAFLD to support more holistic and patient-centered care [4]. The distinction between MASLD and NAFLD is minor, so findings from earlier NAFLD studies can still be considered relevant and applicable to MASLD [5].

Importantly, readers should note that this manuscript uses the term MASLD in alignment with current international recommendations (e.g., AASLD, EASL). However, earlier terminology such as NAFLD or MAFLD is preserved in citations of older studies to maintain accuracy and historical continuity.

Furthermore, NAFLD/MAFLD is closely associated with several endocrine disorders, including hypercortisolism, growth hormone deficiency, and hypothyroidism, reinforcing the intricate relationship between endocrine dysfunction and hepatic steatosis [6,7].

Also, in modern life, environmental pollutants known as endocrine-disrupting chemicals (EDCs) may contribute to the development of metabolic diseases by promoting adipogenesis through the activation of the glucocorticoid receptor (GR), thereby linking EDC exposure to the pathogenesis of obesity and type 2 *diabetes mellitus* [8,9].

The diabetogenic effects of GCs are mediated not only by enhanced hepatic gluconeogenesis and reduced glucose uptake in skeletal muscle, but also by elevated circulating free fatty acids (FFAs), resulting from increased lipolysis in adipose tissue. These metabolic disruptions promote fat accumulation and insulin resistance, creating a self-perpetuating cycle [10].

Obesity and insulin resistance are frequently associated with polycystic ovary syndrome (PCOS), and both are believed to play key roles in its pathogenesis. Increasing evidence suggests that PCOS should be considered not only a reproductive disorder, but also a systemic metabolic disease, associated with an elevated risk of type 2 *diabetes mellitus*, cardiovascular disease, and cerebrovascular complications [11].

Hypothyroidism, both overt and subclinical, is associated with a higher risk of MASLD due to impaired lipid metabolism and liver fat buildup, with the risk especially elevated in younger individuals and men [12].

## 2. Metabolic Impact of Glucocorticoids

GCs play a critical role in enhancing glucose availability to support brain function during periods of acute stress or starvation and can temporarily improve glucose tolerance in such contexts [13]. However, chronic exposure to elevated levels of GCs leads to sustained hyperglycemia and contributes to the development of prediabetes or type 2 *diabetes mellitus*, both characterized by insulin resistance [14]. Persistent hypercortisolemia, as observed in conditions such as Cushing’s syndrome, is well established to promote a constellation of metabolic abnormalities, including hepatic steatosis [15,16].

GCs stimulate lipolysis, muscle protein catabolism, and hepatic gluconeogenesis—interconnected metabolic processes that collectively contribute to systemic insulin resistance. Chronic stress activates the hypothalamic–pituitary–adrenal (HPA) axis, increasing cortisol and catecholamine levels, which in turn promotes visceral adiposity and insulin resistance. These hormonal changes are strongly associated with the development of MASLD [17].

Persistent HPA axis overactivation, common in stress-related disorders, is linked to central fat accumulation and an elevated waist-to-hip ratio. This dysregulation is marked by reduced cortisol variability, a disrupted circadian rhythm, lower morning cortisol levels, and a blunted postprandial cortisol response [18,19].

In the following sections, we will examine how GCs contribute to insulin resistance by altering metabolic pathways, promoting adipose tissue inflammation, and inducing hepatic steatosis. We will also briefly explore their effects on skeletal muscle and pancreatic function, which further exacerbate systemic insulin resistance.

### 2.1. The Effects of Glucocorticoid Hormones on Adipose Tissue

The overall effect of GCs on fat storage is context-dependent, influenced by the metabolic state (Figure 1). Under catabolic conditions, GCs upregulate hormone-sensitive lipase (HSL), promoting lipolysis in adipose tissue and reducing glucose uptake by adipocytes. In contrast, under insulin-rich conditions, GCs enhance lipoprotein lipase (LPL) expression, increasing the hydrolysis of circulating triglycerides from chylomicrons and VLDL. This elevates plasma FFA levels, leading to ectopic fat deposition in the liver, muscle, and central adipose tissue [20,21].

Additionally, GC effects vary by fat depot location (Figure 2). In insulin-resistant states, visceral adipocytes remain sensitive to the synergistic actions of insulin and GCs, maintaining high LPL activity. This favors central fat accumulation, exacerbating abdominal obesity and insulin resistance. Elevated levels of FFAs further drive ectopic lipid accumulation and perpetuate metabolic dysfunction [22,23].

Moreover, GCs may stimulate lipolysis in subcutaneous fat depots while simultaneously promoting differentiation and hypertrophy in visceral fat, a pattern reminiscent of the fat redistribution observed in Cushing’s disease [24,25]. The preferential impact of GCs on visceral adipose tissue has been associated with higher local cortisol production and elevated GR expression [25]. Additionally, human omental adipocytes exhibit approximately twice the glucose uptake rate of subcutaneous adipocytes, likely due to higher GLUT4 expression. However, GCs significantly suppress glucose uptake and downregulate insulin signaling proteins in omental adipocytes, contributing to insulin resistance in visceral fat [26].

Adipocyte differentiation is commonly induced using adipogenic “cocktails” composed of insulin, glucocorticoids, and methylxanthines. These components activate key pathways—insulin-like growth factor (IGF) signaling, glucocorticoid receptor signaling, and cyclic AMP (cAMP) signaling, respectively—all of which play critical roles in the fate determination and maturation of preadipocytes into mature adipocytes [27].

Glucocorticoids play a central role in regulating the gene networks that govern essential functions in human adipose tissue. They upregulate the expression of the enzymes involved in triglyceride hydrolysis—including adipose triglyceride lipase (ATGL), hormone-sensitive lipase (HSL), and monoacylglycerol lipase (MGL)—thereby enhancing lipid turnover [28]. However, in the presence of insulin, simulating a fed state, GCs induce the expression of genes involved in lipid uptake (e.g., lipoprotein lipase), triglyceride synthesis (e.g., 1-acylglycerol-3-phosphate O-acyltransferase and diacylglycerol O-acyltransferase 1), and de novo lipogenesis (e.g., acetyl-CoA carboxylase and fatty acid synthase). Additionally, the GC-induced expression of phosphoenolpyruvate carboxykinase 1 (PEPCK1) likely supports glyceroneogenesis, further contributing to the triglyceride synthesis pathway [29].

As previously highlighted, chronic glucocorticoid (GC) exposure is associated with increased visceral fat accumulation. Transgenic mice with the selective overexpression of 11β-hydroxysteroid dehydrogenase type 1 (11β-HSD1) in adipose tissue develop pronounced visceral obesity, which is further exacerbated by a high-fat diet, along with insulin resistance and diabetes [30]. Although circulating cortisol levels are not consistently elevated in individuals with obesity, the enhanced local conversion of cortisone to cortisol within adipose tissue has been strongly linked to adiposity [23,24].

In individuals with an increased waist circumference, the dysregulation of the HPA axis has been observed, including a loss of diurnal cortisol variation and resistance to suppression in low-dose dexamethasone tests—both indicators of chronic HPA axis hyperactivity [30,31]. Furthermore, elevated urinary free cortisol excretion has been reported in patients with central adiposity [32].

Lee M.J. et al. underscored the importance of circadian GC fluctuations and their interplay with meal timing in modulating the anabolic effects of GCs on adipocytes [24]. Also, it was demonstrated that a cortisol pulse increased leptin levels only when it coincided with food intake or insulin administration [33]. Moreover, GCs suppress adiponectin—a key adipokine that promotes insulin sensitivity—thereby further contributing to metabolic dysfunction [34].

In addition to their effects on white adipose tissue, GCs inhibit the development and activity of brown adipose tissue (BAT), likely through GR-mediated pathways [35].

In Cushing disease, adipose tissue dysfunction and systemic glucocorticoid resistance are prominent features. Interestingly, serum cortisol levels have been positively correlated with irisin, whereas urinary cortisol levels show an inverse relationship with uncoupling protein 1 (UCP-1), a key thermogenic marker in brown adipose tissue [36].

Despite their well-established anti-inflammatory properties, chronic GC exposure—as seen in Cushing syndrome—paradoxically promotes systemic inflammation. This state contributes to glucose intolerance, dyslipidemia, and increased cardiovascular risk. Patients with active or treated Cushing syndrome exhibit elevated inflammatory markers, including C-reactive protein (CRP), interleukin-6 (IL-6), and tumor necrosis factor-alpha (TNF-α) [37]. Elevated circulating FFAs, commonly seen in GC excess, further amplify inflammation through the activation of the NF-κB and JNK signaling pathways, stimulating cytokine production and matrix metalloproteinase (MMP) activity—mechanisms implicated in atherogenesis [38].

### 2.2. The Effects of Glucocorticoid Hormones on Muscles

GCs regulate insulin action in a tissue-specific manner, enhancing insulin sensitivity in subcutaneous adipose tissue under certain conditions, while impairing it in skeletal muscle. This increased insulin responsiveness in adipose tissue, due to the upregulation of lipogenic enzymes, may promote adipocyte expansion and contribute to the development of obesity. In contrast, the reduction in insulin efficacy observed in skeletal muscle appears to result from the GC-mediated inhibition of key intermediary signaling pathways between the insulin receptor and glucose transport activation. Defects in GLUT4 translocation and post-translational modifications that impair the functional capacity of glucose transporter units play an important role [39]. Furthermore, cortisol activates glycogen synthase kinase-3 (GSK-3), a kinase that suppresses glycogen synthesis and promotes protein degradation [40]. Excess glucocorticoids stimulate skeletal muscle protein breakdown, leading to muscle atrophy. The released amino acids are utilized by the liver for gluconeogenesis, contributing to elevated blood glucose levels (Figure 3). Part of this glucose load is subsequently converted into fatty acids and stored as triglycerides, particularly in visceral adipose depots [41].

Elevated plasma free fatty acids (FFAs), a hallmark of glucocorticoid excess, are associated with metabolic disturbances and lipotoxicity in cardiac muscle. When the influx of FFAs exceeds the heart’s oxidative capacity, lipids accumulate and trigger cardiomyocyte damage—either directly or via lipotoxic intermediates such as ceramides. Moreover, glucocorticoid treatment alters the myocardial fatty acid composition by reducing levels of linoleic and γ-linolenic acids while increasing arachidonic acid, which may exert toxic effects on cardiac tissue [42].

### 2.3. The Effects of Glucocorticoid Hormones on Liver

The liver stores triglycerides as a protective mechanism to prevent the accumulation of lipotoxic intermediates such as diacylglycerols and ceramides, which can trigger the formation of reactive oxygen species (ROS). However, even modest increases in the hepatic triglyceride content are associated with insulin resistance [43]. Cortisol plays a crucial role in hepatic lipid metabolism, although its effects vary depending on the physiological context and duration of exposure. Elevated cortisol levels promote hepatic fat accumulation, thereby aggravating conditions such as MASLD [7].

An increased influx of glycerol and free fatty acids to the liver enhances triglyceride synthesis, both through increased substrate availability and the transcriptional activation of lipogenic pathways. Key enzymes involved in glucocorticoid metabolism—such as 11β-hydroxysteroid dehydrogenase types 1 and 2 (11β-HSD1/2) and 5α-/5β-reductases—are primarily expressed in adipocytes and hepatocytes. Moreover, 11β-HSD1 activates glucocorticoids by converting cortisone to cortisol, whereas 11β-HSD2 catalyzes the reverse reaction, inactivating cortisol. Meanwhile, 5α- and 5β-reductases metabolize glucocorticoids into inactive compounds, thereby fine-tuning their local and systemic effects [44].

Experimental studies have shown that the overexpression of 11β-HSD1 exacerbates hepatic steatosis [45], while the pharmacological inhibition of this enzyme improves insulin sensitivity and reduces the hepatic triglyceride content. Similarly, impaired 5α-reductase activity has been associated with worsened steatosis [46].

Notably, hepatic glucocorticoid metabolism appears to shift during disease progression—from simple steatosis to steatohepatitis—suggesting an adaptive enzymatic response to inflammation (Figure 4). In simple steatosis, 5α-reductase levels increase while 11β-HSD1 expression decreases, resulting in reduced intrahepatic cortisol levels. In contrast, during steatohepatitis, 5α-reductase levels decline and 11β-HSD1 expression rises, leading to elevated hepatic cortisol concentrations [47,48].

In human patients, chronic insulin resistance and a reduced mitochondrial oxidative capacity create a permissive environment for the rapid progression from steatosis to steatohepatitis and ultimately to cirrhosis [7].

A significant association has been identified between various insulin resistance (IR) indices and liver fibrosis in patients with NAFLD, underscoring the clinical utility of IR markers in routine care to identify high-risk individuals and implement early interventions aimed at preventing fibrosis progression and improving outcomes [49]. (Note: Although the original study refers to NAFLD, the current terminology adopted in this review is MASLD.)

Under physiological conditions, FFAs released from white adipose tissue (WAT) activate hepatic peroxisome proliferator-activated receptor alpha (PPARα), enhancing fatty acid oxidation and attenuating inflammation. Adiponectin, an anti-inflammatory adipokine, combats hepatic steatosis and inflammation by activating both PPARα and AMP-activated protein kinase (AMPK). Additionally, hepatic PPARα controls circulating levels of fibroblast growth factor 21 (FGF21), a hormone that alleviates hepatic lipid accumulation partly by stimulating brown adipose tissue activity [50,51].

Glucocorticoids, such as dexamethasone and insulin, exhibit complex interactions in the regulation of the hepatic gene expression involved in glucose and lipid metabolism. In primary hepatocytes isolated from Sprague Dawley rats, dexamethasone has been shown to potentiate the insulin-induced expression of sterol regulatory element-binding protein-1c (SREBP-1c), while suppressing phosphoenolpyruvate carboxykinase (Pck1) expression in a time-dependent manner [52]. Dexamethasone-induced hyperinsulinemia is also associated with reduced insulin clearance, attributed to diminished hepatic insulin-degrading enzyme (IDE) activity. In glucocorticoid-treated rats, this reduction in IDE activity may contribute to compensatory hyperinsulinemia. These findings suggest that the partial or short-term inhibition of IDE may offer a therapeutic avenue to improve glycemic control [53].

Glucocorticoids promote gluconeogenesis in both the liver and kidneys. Upon glucocorticoid stimulation, the expression of Kruppel-like factor 9 (KLF9) is induced, enhancing the recruitment of transcriptional coactivators and the binding of transcription factors to glucocorticoid response elements within gluconeogenic gene promoters. This pathway amplifies the transcriptional activation of phosphoenolpyruvate carboxykinase (PEPCK) and glucose-6-phosphatase (G6Pase), thereby promoting gluconeogenesis [35,54].

Dexamethasone also induces insulin resistance by inhibiting hepatic hexokinase activity, suppressing hepatic glucose oxidation and stimulating hepatic lipogenesis, ultimately leading to lipid accumulation in the liver. Furthermore, it contributes to dyslipidemia by increasing the expression of lipogenic enzymes, by decreasing lecithin–cholesterol acyltransferase (LCAT) activity and elevating free cholesterol levels [41,55].

### 2.4. The Effects of Glucocorticoid Hormones on the Pancreas

GCs, such as cortisol, impair pancreatic β-cell function by binding to glucocorticoid receptors, thereby reducing insulin gene transcription and secretion. They also decrease the production of glucagon-like peptide-1 (GLP-1), diminishing its insulinotropic effects, and increase somatostatin secretion, which further suppresses insulin synthesis and release [40].

GCs can directly induce apoptosis in pancreatic β-cells by upregulating pro-apoptotic proteins (e.g., Bax, BAD, p38), downregulating anti-apoptotic signals (e.g., Bcl-2), and inhibiting cell survival pathways such as the cAMP–PKA axis [56]. This pro-apoptotic shift, coupled with heightened sensitivity to oxidative stress and metabolic insults, leads to β-cell loss despite an initial compensatory proliferation phase [57]. Pancreatic beta cells are extremely sensitive to oxidative stress. Intracellular oxidative stress, characterized by the downregulation of antioxidant enzymes and upregulation of pro-oxidant enzymes like NADPH oxidase, contributes significantly to β-cell dysfunction [58]. Additionally, endoplasmic reticulum (ER) stress—mediated by factors such as activating transcription factor 6 (ATF6), C/EBP homologous protein (CHOP), and Bcl-2 suppression—further exacerbates glucocorticoid-induced β-cell apoptosis [59].

GC therapy also downregulates the expression of key β-cell genes, including the long intergenic non-coding RNA GAS5 (Growth Arrest-Specific 5), which plays a pivotal role in regulating apoptosis, cell proliferation, and cycle arrest. Restoring GAS5 expression has been shown to mitigate glucocorticoid-induced β-cell damage, suggesting its potential as a therapeutic target to protect β-cell integrity [60].

### 2.5. Some Practical Applications Based on the Link Between Glucocorticoids and Insulin Resistance

Kumar et al. [61] proposed that high doses of dexamethasone (16 mg/kg intraperitoneally for six days) may serve as an effective animal model for investigating the secondary effects of insulin resistance. This treatment induced hepatic steatosis and mild to moderate arteriosclerosis in Wistar rats, likely through mechanisms involving GC-induced insulin resistance. Other studies have also confirmed that dexamethasone administration leads to significant liver injury, making it a valuable model for exploring hepatic disease pathogenesis and potential therapeutic strategies. Experimental findings include elevated fasting blood glucose, increased serum ALT and AST levels, and histopathological changes such as hydropic degeneration, portal edema, leukocyte infiltration, and fibrosis [62].

In terms of therapeutic interventions, SH-01D, an aqueous leaf extract of *Melia azadirachta* (neem), demonstrated significant antidiabetic effects in fructose-fed and dexamethasone-treated rats. At a dose of 60 mg/kg, it lowered the plasma glucose, insulin, triglyceride, and LDL levels, suggesting its potential for mitigating insulin resistance. However, further studies are required to evaluate its toxicity and safety profile [63]. Similarly, coriander oil was shown to alleviate dexamethasone-induced oxidative stress, hyperglycemia, and dyslipidemia by reducing blood glucose and HbA1c levels, enhancing insulin secretion, and protecting pancreatic islets—supporting its potential use as an adjuvant in the management of type 2 diabetes [64].

In clinical settings, treatment with the glucocorticoid receptor antagonist mifepristone in patients with hypercortisolism has led to improvements in both biochemical markers and imaging findings associated with NAFLD [65]. (Note: The study cited refers to NAFLD; this review adopts the updated term MASLD). Reductions in liver enzyme levels and the resolution of hepatic steatosis were observed, highlighting the therapeutic potential of glucocorticoid receptor antagonism in MASLD associated with hypercortisolism.

Moreover, combining the Mediterranean diet with moderate aerobic exercise has been shown to significantly reduce liver fat accumulation and circulating cortisol levels. The most pronounced effects on cortisol were observed in individuals undergoing high-intensity interval training, emphasizing the beneficial role of structured exercise in modulating both hepatic lipid content and glucocorticoid status [66]. Importantly, the authors noted that mid-adolescence may represent a critical window for interrupting the fat mass–insulin resistance cycle, potentially lowering the risk of long-term metabolic complications [67].

Recently, researchers developed a human-based pancreas–liver microphysiological system that replicates glucocorticoid-induced diabetes. This in vitro model recapitulates key features of metabolic dysfunction and offers a promising platform for drug discovery and mechanistic research [68].

Considering the weight of the current evidence and the consistent association between elevated glucocorticoid levels and the features of metabolic syndrome, a strong connection between glucocorticoid excess and MASLD is highly likely—an association that clinicians should recognize in both diagnosis and management.

## 3. Sexual Hormones and Metabolic Impact

Polycystic ovary syndrome (PCOS) is one of the most common endocrine disorders affecting women of a reproductive age. Based on its classical definition, the condition is characterized by ovulatory dysfunction, hyperandrogenism, and/or polycystic ovarian morphology, as confirmed by ultrasonography. Ovulatory dysfunction is clinically represented by oligo- or amenorrhea and subfertility, while hyperandrogenism typically manifests as hirsutism, acne, or alopecia [69].

Obesity and insulin resistance are frequently associated with PCOS and are believed to play key roles in its pathogenesis [70]. Another condition closely linked to insulin resistance is MASLD, one of the leading causes of chronic liver disease in Western countries. PCOS and NAFLD share common risk factors such as type 2 *diabetes mellitus*, obesity, dyslipidemia, and genetic predisposition, with insulin resistance acting as a central mechanism in each [71].

Women with PCOS, particularly those with hyperandrogenemia, obesity, and insulin resistance, exhibit a higher prevalence of MASLD, suggesting that PCOS and MASLD are pathophysiologically linked rather than coincidentally [72].

### 3.1. PCOS and MASLD

The pathophysiology of PCOS involves an intrinsically high frequency of gonadotropin-releasing hormone (GnRH) pulses along with reduced hypothalamic negative feedback from ovarian sex steroids, likely due to a deficiency of progesterone receptors in hypothalamic neurons [73]. GnRH pulsatility in the infundibular nucleus favors luteinizing hormone (LH) secretion while suppressing follicle-stimulating hormone (FSH) release. In individuals with PCOS, elevated LH levels stimulate ovarian thecal cells to overproduce androgens, while insufficient FSH contributes to anovulation and the characteristic polycystic ovarian morphology [74].

Historically, PCOS was described as a disorder of ovarian androgen excess. However, it is now recognized that adrenal and peripheral tissues also significantly contribute to androgen overproduction [75]. Approximately 30% of women with PCOS exhibit signs of adrenal hyperactivity, reflected in the elevated plasma levels of dehydroepiandrosterone sulfate (DHEAS). This is thought to result from intrinsic hyperactivity of the adrenal cortex rather than pituitary or hypothalamic dysfunction [76].

Hyperandrogenemia promotes visceral fat accumulation, exacerbating insulin resistance and hyperinsulinemia. Insulin acts synergistically with LH to further enhance androgen production in both the ovaries and adrenal glands. In addition, insulin suppresses hepatic sex hormone-binding globulin (SHBG) synthesis, increasing free (bioactive) androgen levels, and impairs progesterone-mediated negative feedback on GnRH neurons—thereby perpetuating the cycle of androgen excess [77].

The role of insulin resistance in MASLD among women with PCOS is supported by studies showing that improving insulin sensitivity—through weight loss or pharmacologic agents—ameliorates reproductive, metabolic, and hyperandrogenic symptoms. Insulin resistance increases lipolysis in adipose tissue, releasing excessive FFAs into circulation. These FFAs are taken up by the liver, promoting intrahepatic lipid accumulation and contributing to MASLD development [78].

Hyperandrogenism also appears to contribute to MASLD through the downregulation of low-density lipoprotein receptor (LDLR) expression in hepatocytes. Androgens reduce LDLR gene expression, making it harder for the liver to clear plasma LDL cholesterol. As a compensatory response, the liver may upregulate the synthesis of very low-density lipoprotein (VLDL), further exacerbating hepatic fat accumulation. In contrast, estrogens normally upregulate LDLR expression via estrogen-responsive elements in hepatocytes. This opposing regulation suggests a direct link between hyperandrogenism and LDLR downregulation in PCOS, potentially increasing the risk of MASLD [79].

Central obesity and adipose tissue dysfunction are additional contributors to MASLD in PCOS. Women with PCOS tend to have hypertrophic adipocytes—larger fat cells—rather than hyperplastic expansion (an increased number), and hypertrophic adipose tissue is closely associated with insulin resistance due to reduced insulin sensitivity at the cellular level [80].

Other mechanisms involved in MASLD pathogenesis in PCOS include dysregulated adipocytokine secretion and genetic predisposition (Figure 5). For instance, adiponectin—an anti-inflammatory, insulin-sensitizing adipokine—is the only known adipokine consistently downregulated in obesity. Women with PCOS have lower adiponectin levels, which are associated with increased gluconeogenesis, enhanced fatty acid synthesis in hepatocytes, and subsequent hepatic steatosis [81].

Several genetic polymorphisms have been implicated in both PCOS and MASLD. These include the LDLR gene (affecting LDL receptor expression in hepatocytes), the PNPLA3 gene (encoding a protein strongly linked to hepatic fat accumulation), and the FTO gene (associated with obesity and altered adipose tissue biology). However, in women with PCOS, metabolic factors outweigh PNPLA3 polymorphism in determining NAFLD risk, though both can synergistically increase disease severity [81].

### 3.2. Estrogen Deficiency and MAFLD in Menopause

Estrogen has been shown to play a key role in NAFLD, as evidenced by comparisons between premenopausal women and those with PCOS, or postmenopausal women [82]. Targeting hyperandrogenism and insulin resistance in PCOS may improve the clinical profile via estrogen modulation, which impacts NAFLD through several mechanisms. Estrogen enhances the hepatic β-oxidation of fatty acids and reduces de novo lipogenesis, thereby limiting fat accumulation in the liver [83]. It improves the lipid profile by increasing HDL cholesterol and reducing LDL and triglycerides, contributing to decreased hepatic fat storage [84]. Estrogen also reduces liver inflammation by modulating Kupffer cell activity and inhibiting the NF-κB signaling pathway, which is central to cytokine production. Furthermore, it decreases liver fibrosis by inhibiting the activation of hepatic stellate cells, inhibiting NF-κB signaling [85], and downregulating pro-fibrotic markers such as TGF-β and collagen types I and III [86]. The prevalence and severity of metabolic dysfunction-associated steatotic liver disease (MASLD) increase after menopause, likely due to hormonal changes affecting fat distribution and metabolism. While animal studies suggest the protective role of estrogens, clinical evidence on the impact of menopausal hormone therapy remains limited and inconsistent [87]. The risk of developing MAFLD was higher in women with premature menopause (<40 years) than in those with menopause aged ≥ 50 years [88]. Some researchers believe that hormone replacement therapy (HRT) could be a protective factor against this liver disease [89]. Metabolomics offers a promising approach to identify molecular profiles, disease severity, and sex-specific biomarkers [90]. Importantly, the sex-specific identification of biomarkers holds significant promise for improving risk stratification and personalized management in MASLD. Techniques such as metabolomics can uncover distinct molecular signatures between men and women, offering valuable diagnostic and prognostic tools tailored to sex differences [90].

### 3.3. Therapeutic Strategies Targeting Insulin Resistance and Hepatic Steatosis in PCOS

Lifestyle modifications—including diet, weight loss, and increased physical activity—either alone or in combination with pharmacotherapy, have demonstrated clear benefits in individuals with both PCOS and hepatic steatosis. Treatment strategies aim to address the key metabolic risk factors of metabolic dysfunction-associated steatotic liver disease (MASLD) in women with PCOS, including insulin resistance, prediabetes, and dyslipidemia. Commonly used medications include insulin sensitizers (such as metformin and pioglitazone), lipid-lowering agents, and liver-protective compounds such as antioxidants and anti-inflammatory drugs.

Metformin is an effective adjunct to lifestyle interventions for managing polycystic ovary syndrome in adults, particularly those with an elevated body mass index, insulin resistance, and dyslipidemia. It reduces hepatic gluconeogenesis, improves insulin sensitivity, and has been shown to enhance liver function and mitigate metabolic syndrome symptoms in women with PCOS [91,92].

A retrospective study of 113 women with polycystic ovary syndrome assessed metformin’s effects in insulin-resistant and non-insulin-resistant groups. Metformin improved menstrual regularity and ovulation in both, reduced free androgen index levels, and improved metabolism in insulin-resistant patients. Improvements in non-insulin-resistant patients suggest benefits beyond insulin resistance. The study’s large sample size and consistent evaluation by the same doctor enhance its reliability. The limitations of the study include the short three-month treatment period and the focus on infertile patients seeking short-term conception [93].

A recent comprehensive analysis of metformin use in women with polycystic ovary syndrome, including adolescents and adults with or without obesity, shows significant benefits in anthropometric and metabolic outcomes compared to the placebo. Metformin is recommended for adults with a body mass index ≥ 25 kg/m^2^ and may be considered for those with lower BMI and adolescents, though evidence is more limited. Gastrointestinal side effects are generally mild, dose-dependent, and short-term. Despite being off-label, metformin use in PCOS is well supported by evidence [92].

In an observational study it was demonstrated that metformin may have had a beneficial effect on NAFLD in normal and overweight women with PCOS. The study had limitations including small group sizes, a short follow-up period, and reliance on non-imaging biomarkers for fatty liver assessment. Future research should use imaging methods alongside laboratory tests to better evaluate liver health in women with PCOS. Emerging treatments like GLP-1 analogs show promise for improving fatty liver disease, but their effects in PCOS require further study [94].

Thiazolidinediones (TZDs) improve insulin sensitivity through the activation of peroxisome proliferator-activated receptor gamma (PPARγ), and by increasing adiponectin levels, they reduce hepatic gluconeogenesis and fatty acid synthesis. In addition, TZDs promote the redistribution of fat storage toward adipose tissue, thereby decreasing hepatic lipid accumulation. However, the long-term use of TZDs may lead to weight gain and relapse after therapy discontinuation [91].

Glucagon-like peptide-1 (GLP-1) receptor agonists, such as liraglutide, have shown efficacy in improving liver fibrosis markers in obese women with PCOS and MASLD. Their benefits may result from enhanced insulin sensitivity, glucose-dependent insulin secretion, and clinically significant weight loss [95].

Antioxidants, such as vitamin E, may protect hepatocytes from oxidative damage by scavenging reactive oxygen species and inhibiting lipid peroxidation. Vitamin E significantly improves liver function and reduces inflammation in NAFLD/MASLD patients, though its effect on fibrosis resolution remains unclear [96].

Omega-3 fatty acids also offer therapeutic potential in MASLD. Their ability to reduce hepatic triglyceride synthesis, along with their anti-inflammatory and antioxidant properties, contributes to improved lipid and glucose metabolism. Omega-3s enhance lipoprotein lipase activity, modulate hepatic gene expression, and reduce postprandial lipemia. Supplementation has been associated with reductions in HOMA-IR scores, fasting insulin levels, hepatic fat content, and plasma triglycerides in women with PCOS and steatosis. However, omega-3 fatty acids are not considered first-line therapy unless hypertriglyceridemia is also present [97].

## 4. Thyroid Hormones and Metabolic Impact

Thyroid hormones (THs) include T4 (thyroxine) and T3 (triiodothyronine). Although T4 is more abundant in circulation, T3 is the biologically active form and exerts the majority of metabolic effects, including the regulation of lipids, glucose, and energy metabolism [98].

THs play key roles in regulating metabolism, including the basal metabolic rate (BMR), fat and glucose metabolism, and energy balance, with effects varying across tissues depending on the nutritional and hormonal status. These hormones influence multiple organs—such as liver, muscle, and adipose tissue—directly and indirectly, promoting processes like β-oxidation, gluconeogenesis, and thermogenesis under different metabolic states. Traditionally, thyroid disorders were managed by adjusting TH levels based on serum free T4 (FT4) and TSH, but it is now recognized that TH actions can differ by tissue [99].

### 4.1. Thyroid Hormones and Lipid Metabolism

THs stimulate lipolysis in white adipose tissue and mobilize fatty acids. This leads to the generation of free fatty acids, which are transported into hepatic cells via specific protein transporters. FFAs can also derive from “de novo” lipogenesis and the hydrolysis of dietary triglycerides [100]. “De novo” lipogenesis is the metabolic process by which glucose is converted into fatty acids, and it is tightly regulated—activated in the fed state and suppressed during fasting. THs, particularly T3, promote de novo lipogenesis both directly and indirectly. T3 stimulates the transcription of key enzymes such as acetyl-CoA carboxylase (ACC) [101] and fatty acid synthase (FAS) [102], with thyroid response elements identified in their promoter regions. Also, T3 indirectly promotes lipogenesis by increasing the transcription of malic enzyme, which generates NADPH, a key reducing agent required for fatty acid synthesis [103]. Additionally, THs activate the transcription factors as carbohydrate response element-binding protein (ChREBP) [104].

THs play a central role in hepatic cholesterol metabolism by regulating cholesterol synthesis, uptake via low-density lipoprotein receptors (LDL-Rs), and clearance through reverse cholesterol transport. THs strongly influence the expression of HMG-CoA reductase, the rate-limiting enzyme in cholesterol synthesis, more than other hormones such as insulin, estrogen, glucagon, or glucocorticoids [105]. They upregulate LDL-R expression both directly—through binding to TH response elements in the LDL-R promoter—and indirectly via the activation of sterol regulatory element-binding protein (SREBP2), a transcription factor that enhances LDL-R transcription [106].

THs enhance bile acid excretion and thereby influence cholesterol metabolism. T3 upregulates the expression of cholesterol 7α-hydroxylase (CYP7A1), the rate-limiting enzyme in bile acid synthesis, promoting the conversion of cholesterol into bile acids as part of the reverse cholesterol transport pathway [107,108].

Fatty acid β-oxidation is the process of breaking down long-chain fatty acids for energy, especially during energy deficit. THs support fatty acid β-oxidation by restoring hepatic lipase activity, promoting lipophagy to release free fatty acids, and enabling their transport into mitochondria [109]. THs regulate key fatty acid β-oxidation enzymes such as Carnitine palmitoyltransferase 1 alpha (Cpt1α) and enhance mitochondrial function via the electron transport chain and Krebs cycle. THs also promote mitochondrial biogenesis and the recycling of damaged mitochondria (mitophagy) by activating PGC-1α (peroxisome proliferator-activated receptor gamma coactivator 1-alpha) and ERRα (estrogen-related receptor alpha), ensuring efficient energy metabolism [110].

Given these effects, patients with hypothyroidism often present with abnormal blood lipid levels (dyslipidemia), including increased levels of low-density lipoprotein cholesterol, triglycerides, and apolipoprotein B [111]. Moreover, a mutual relationship between abnormal lipid levels and reduced sensitivity to insulin (insulin resistance) has also been demonstrated [112].

### 4.2. TH Dysfunction, Insulin Resistance, and MAFLD

THs are key regulators not only of lipid metabolism, but also of carbohydrate metabolism. They inhibit glycolysis in brown adipose tissue, the heart, and the liver, while promoting hepatic gluconeogenesis and ketogenesis [113]. THs influence glucose metabolism through both the direct transcriptional regulation of target genes and indirect modulation via the hypothalamus and sympathetic nervous system. Notably, they may act as insulin agonists in skeletal muscle and antagonists in the liver. In hyperthyroidism, this imbalance often results in hepatic insulin resistance and impaired glucose tolerance. In contrast, in hypothyroidism, insulin resistance tends to predominate in peripheral tissues, potentially due to impaired mitochondrial function and reduced blood flow to muscle and adipose tissue [114].

In the liver, T3 directly regulates the genes critical for glucose homeostasis. It upregulates glucose-6-phosphatase [115], phosphoenolpyruvate carboxykinase (PEPCK), pyruvate carboxylase (PC) [116], and the β2-adrenergic receptor [117], thereby enhancing gluconeogenesis and glycogenolysis. Conversely, T3 suppresses the expression of Akt2 and Gi proteins, promoting hepatic glucose output and inhibiting glycogen synthesis [100]. Furthermore, T3 increases the expression of GLUT2, facilitating glucose export from the liver [118].

In skeletal muscle and adipose tissue, T3 stimulates GLUT1 and GLUT4 expression, promoting both basal and insulin-stimulated glucose uptake [119]. These tissue-specific actions underscore the complex role of THs in maintaining metabolic flexibility.

In hypothyroid states, decreased hepatic lipase activity and impaired triglyceride clearance contribute to fat accumulation in the liver, a process exacerbated by concurrent insulin resistance [120]. In patients with MASLD, the impaired deiodination of T4 to active T3 may disrupt hepatic TH signaling, further contributing to lipid accumulation and hepatic inflammation [121]. Oxidative stress, dysregulated adipokine profiles, and autoimmune mechanisms—especially in the context of Hashimoto’s thyroiditis—may also contribute to MASLD pathogenesis (Figure 6) [122]. Leptin levels are positively correlated with TSH concentrations and are often elevated in hypothyroid individuals [123]. TSH has been shown to suppress both insulin synthesis and secretion from pancreatic β-cells, thereby increasing blood glucose levels [124]. Accordingly, elevated TSH and low free T4 levels have been associated with an increased risk of type 2 *diabetes mellitus* and a higher likelihood of progression from prediabetes to overt diabetes. In contrast, individuals with high or high–normal thyroid function appear to be protected against the development or progression of type 2 *diabetes mellitus* [125].

### 4.3. MAFLD and Hypothyroidism: An Overlooked Endocrine Link

Recent meta-analyses have demonstrated that primary hypothyroidism is significantly associated with both an increased prevalence and greater histological severity of MASLD [126].

Not only overt hypothyroidism, but also subclinical hypothyroidism (serum TSH concentration above the upper limit of the reference range, but with a normal serum-free T4 concentration) and low–normal thyroid function (normal serum-free T4 concentration and a TSH concentration at a high value within the normal range) are associated with MASLD [127].

In patients with subclinical hypothyroidism, long-term levothyroxine (LT4) therapy may be beneficial in managing NAFLD. While the overall effect in mild subclinical hypothyroidism was modest, significant improvements were observed in those with coexisting dyslipidemia. Additionally, LT4 treatment was associated with a trend toward higher NAFLD remission rates [128].

In fact, the relationship between low thyroid function and MASLD appears to be bidirectional [129,130].

Although the direction of causality is not yet fully established, it is prudent to evaluate thyroid function in MASLD patients and to monitor MASLD risk in individuals with hypothyroidism or subclinical hypothyroidism, particularly if they are overweight or obese [131]. The early diagnosis and monitoring of MASLD are essential, as low–normal or subclinical hypothyroidism significantly increases all-cause and cardiovascular mortality risk (Y.L. Chen) [132].

### 4.4. Therapeutic Strategies Targeting THR

The connection between thyroid dysfunction and liver fibrosis may involve excessive extracellular matrix deposition, hepatic stellate cell activation, increased collagen synthesis, and impaired collagen degradation [133,134].

Experimental studies have demonstrated that thyroid hormone receptor (THR) agonists can reduce hepatic steatosis [135,136]. A nuanced understanding of TR isoform-specific actions thus represents a critical step toward the development of targeted, organ-selective therapies for metabolic disorders such as MAFLD and NASH.

T3, the active form of THs, exerts many of its actions through its receptors (TRs): TRα1, TRβ1, and TRβ2. These THR isoforms exhibit distinct tissue distributions and functions, influencing endocrine regulation and therapeutic targeting. TRα1 is mainly expressed in the brain, heart, and skeletal muscle, where it regulates neurodevelopment, cardiac function, and thermogenesis, and may serve as a target for cardiac regeneration. TRβ1 is found predominantly in the liver, kidney, and thyroid, while TRβ2 is localized to the retina, cochlea, and pituitary. Despite structural similarity, TRα1 and TRβ1 differ in cofactor recruitment and target gene regulation. TRα2, although abundant in the brain, does not bind T3 and lacks classical receptor activity [137,138,139].

TRβ1 has emerged as a primary target for TH-based pharmacotherapy in MAFLD due to its high hepatic expression and central role in lipid metabolism, cholesterol homeostasis, and glucose regulation. This focus has shifted to TRβ1-selective thyromimetics for their dual ability to reduce plasma cholesterol and enhance reverse cholesterol transport [140].

TRα and TRβ differ by only a single amino acid, which has enabled the development of TRβ1-selective thyromimetics aimed at minimizing cardiovascular side effects. The primary effect of TRβ agonists is to enhance hepatic β-oxidation, thereby reducing hepatic lipid accumulation [141,142,143].

Patients with TRβ mutations often exhibit elevated TH levels, an increased metabolic rate, and cardiac hyperthyroidism, yet show hepatic resistance, as evidenced by dyslipidemia (elevated LDL-c, TG, and reduced HDL-c) compared to controls. These findings support monitoring lipid profiles and hepatic fat content, especially in those patients with deleterious TRβ variants, and suggest initiating lipid-lowering therapy for hypercholesterolemia [144]. Thus far, only a few patients with mutations in TRα1 have been identified: their TH levels have been near-normal, but they have retarded growth, skeletal dysplasia, a reduced metabolic rate, and cardiac hypothyroidism [145].

TRβ agonists represent a promising class of therapeutics for metabolic liver diseases due to their liver-selective action and lipid-lowering effects. Several TRβ agonists have been included in clinical studies. Resmetirom (MGL-3196) is a liver-targeted TRβ agonist that reduces liver fat, cholesterol, and triglycerides, while also inhibiting hepatic steatosis and fibrosis and offering cardioprotective benefits. Its FDA approval was supported by Phase 3 trial data demonstrating significantly greater rates of NASH resolution and fibrosis improvement versus placebo, establishing it as the first approved treatment for active fibrotic steatohepatitis [146,147].

Resmetirom trials show promising effects on hepatic steatosis and fibrosis in MASLD, yet limitations exist. They primarily use surrogate endpoints rather than clinical outcomes like cirrhosis progression. Strict exclusion criteria omit patients with advanced liver disease, diabetes, and cardiovascular comorbidities, limiting generalizability. Trial adherence and tolerability are optimized through close monitoring, unlike real-world settings where gastro-intestinal side effects and fasting requirements may reduce compliance [148].

Also, other THR-β agonists have been developed. Clinical trials have demonstrated that MB07811 (VK2809) significantly reduces liver fat content and LDL-C levels, while exhibiting a promising safety and tolerability profile. It is currently undergoing evaluation in a Phase IIb trial [131,149].

Eprotirome (KB2115) is a liver-targeted TRβ agonist with triglyceride-lowering effects, halted in Phase III due to liver enzyme elevations and cartilage damage seen in long-term dog studies [150,151].

Studies with Sobetirome (GC-1), a TRβ-selective agonist known for its lipid-lowering and weight-loss effects, were halted in Phase 1 due to early-onset hyperglycemia and insulin resistance [146,152].

Experimental and clinical investigations of thyromimetics have demonstrated both common mechanisms of action and distinct variations in therapeutic outcomes and safety profiles, reflecting the complexity of translating these agents from preclinical studies to clinical use [153]. The relationship between thyroid hormones (THs) and MASLD appears to be modulated not only by factors such as obesity, insulin resistance, and other components of metabolic syndrome, but may also differ across MASLD phenotypes, highlighting the complexity of the studies that need to be conducted [154]. Moreover, this year, the U.S. Food and Drug Administration (FDA) released a strategic roadmap aimed at replacing animal testing in preclinical safety evaluations with human-relevant technologies, including organ-on-chip systems, advanced cell culture methods, and artificial intelligence. This initiative marks a significant advancement in biomedical research [155].

## 5. Hormonal Crosstalk, Inflammation, and Immune Modulation

As previously discussed, individual hormones such as glucocorticoids, sex hormones, insulin, and thyroid hormones exert distinct metabolic effects. However, their interplay in modulating inflammation and immune responses constitutes a critical link in the pathogenesis of metabolic disorders such as MASLD. This crosstalk is summarized below.

Inflammation and immune response represent crucial links between hormonal dysregulation, adipose tissue dysfunction, and metabolic disease. Glucocorticoids (GCs) exhibit potent immunosuppressive and anti-inflammatory effects via the inhibition of the NF-κB and AP-1 signaling pathways, leading to the decreased expression of pro-inflammatory cytokines such as IL-6, TNF-α, and CRP [37,38]. However, chronic GC exposure paradoxically promotes systemic low-grade inflammation, partly through lipotoxicity and increased free fatty acid release, which activate innate immune pathways and contribute to insulin resistance and vascular dysfunction [23,38,41].

Sex steroid hormones also modulate immune responses: estrogens generally exert anti-inflammatory effects by reducing TNF-α and promoting regulatory T-cell function [85], whereas androgens often enhance pro-inflammatory cytokine production and contribute to adipose tissue hypertrophy and fibrosis [77,80]. Thyroid hormones (THs), particularly triiodothyronine (T3), play immunomodulatory roles by influencing cytokine production, oxidative stress, and macrophage polarization. Hypothyroidism is associated with a pro-inflammatory state and increased levels of IL-6, TNF-α, and CRP, which may exacerbate hepatic inflammation and insulin resistance. Conversely, hyperthyroidism can contribute to immune dysregulation through enhanced autoimmunity and increased oxidative stress [7,121,123].

Insulin and adipose tissue further influence immune homeostasis; insulin resistance fosters a pro-inflammatory milieu characterized by M1 macrophage infiltration into visceral fat depots, elevated IL-1β, and reduced adiponectin [23,80,81]. Adipose tissue acts as an endocrine organ, secreting adipokines that modulate both local and systemic inflammation [23,80]. It should also be emphasized that lipid overload induces mitochondrial dysfunction, leading to excessive ROS production and oxidative stress. This oxidative stress causes the release of mitochondrial DNA into the cytoplasm, which is recognized as a damage-associated molecular pattern that activates immune responses and amplifies inflammation. In parallel, ROS activate inflammasome signaling pathways that promote further inflammation through cytokines, which impair mitophagy, worsen mitochondrial dysfunction, and perpetuate a vicious cycle of oxidative stress, inflammation, and immune activation that accelerates MASLD progression [156,157]. Hormone–adipose–immune interactions contribute to MASLD and endocrine disorders, with inflammation acting as both a cause and effect of hormonal imbalance [7,80,85].

## 6. Future Directions: Human-Relevant Models and Precision Medicine

The future of research into hormone-driven MASLD lies in the integration of innovative technologies that bridge the gap between molecular insights and clinical applications. Organ-on-chip systems, including liver-on-chip and adipose-tissue-on-chip platforms, replicate human microarchitecture, perfusion, and hormone responsiveness, enabling sophisticated studies of the interactions between glucocorticoids, sex hormones, insulin, thyroid hormones, and adipokines in MASLD pathogenesis [158]. Unlike static cell cultures or animal models, organ-on-chip devices can simulate dynamic hormonal fluctuations, inflammatory signaling, and metabolic stressors, thereby capturing sex-specific and tissue-specific responses with unprecedented precision [159]. Recent studies have successfully modeled steatosis, inflammation, and fibrogenesis on liver-on-chip platforms, demonstrating these systems’ potential for drug screening, toxicity testing, and biomarker discovery in MASLD research [160]. Moreover, artificial intelligence (AI) and machine learning approaches are emerging as pivotal tools for deciphering large-scale multi-omics datasets, including transcriptomics, proteomics, and metabolomics, to predict disease trajectories and stratify patient risk profiles [161]. AI-driven analyses of metabolomic data have already identified sex-specific lipid signatures associated with steatosis severity and insulin resistance, offering novel pathways for personalized diagnostics and therapeutic interventions [162]. The integration of AI analytics with organ-on-chip technologies holds exceptional promise for virtual drug screening, precision biomarker discovery, and individualized therapeutic approaches for MASLD. Nonetheless, significant challenges remain, including data standardization, model validation across diverse populations, and the need for robust translational pipelines to ensure that findings from organ-on-chip systems and AI analyses effectively translate into clinical practice [163,164]. Despite these hurdles, the convergence of these advanced technologies represents a transformative step toward precision hepatology and the personalized management of endocrine-driven liver diseases.

## 7. Conclusions

Metabolic imbalances caused by hormonal dysfunctions represent a major challenge for clinicians and researchers when interpreting diagnostic results and developing effective therapies. Beyond integrated clinical care, advancing our understanding of hormone-driven MASLD pathways demands innovative research methodologies. Human-relevant platforms such as organ-on-chip systems and artificial intelligence-based models are not merely optional tools, but essential technologies for the precise simulation, validation, and translation of complex endocrine mechanisms into clinical practice. These advanced approaches will accelerate biomarker discovery—including sex-specific signatures—and optimize therapeutic strategies, ultimately bridging the gap between mechanistic insights and patient care. Therefore, future multidisciplinary efforts should prioritize the integration of organ-on-chip technologies and AI-driven analytics alongside traditional clinical collaboration to achieve meaningful progress in the diagnosis, treatment, and prevention of hormone-related metabolic diseases. Particular interest has been focused on the development of thyromimetic compounds targeting TRβ1, not only because of their ability to lower plasma cholesterol but also due their ability to stimulate RCT.

## Figures and Tables

**Figure 1 cimb-47-00565-f001:**
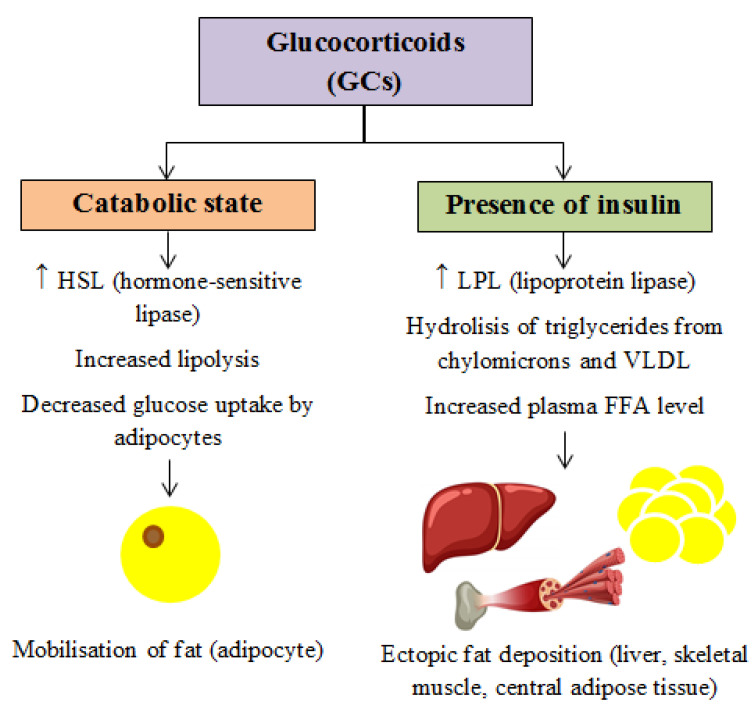
Effects of GCs depending on the metabolic state.

**Figure 2 cimb-47-00565-f002:**
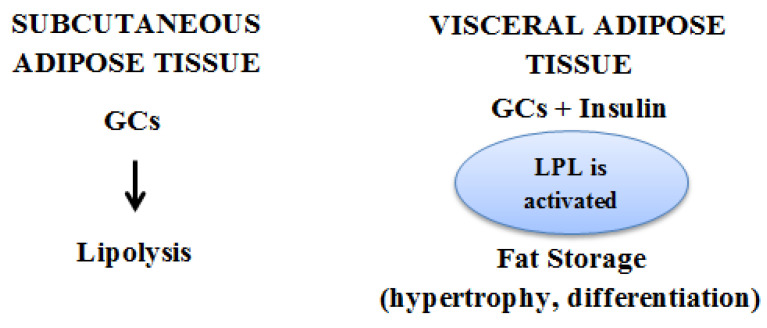
Different effects of GCs on subcutaneous and visceral adipose tissue.

**Figure 3 cimb-47-00565-f003:**
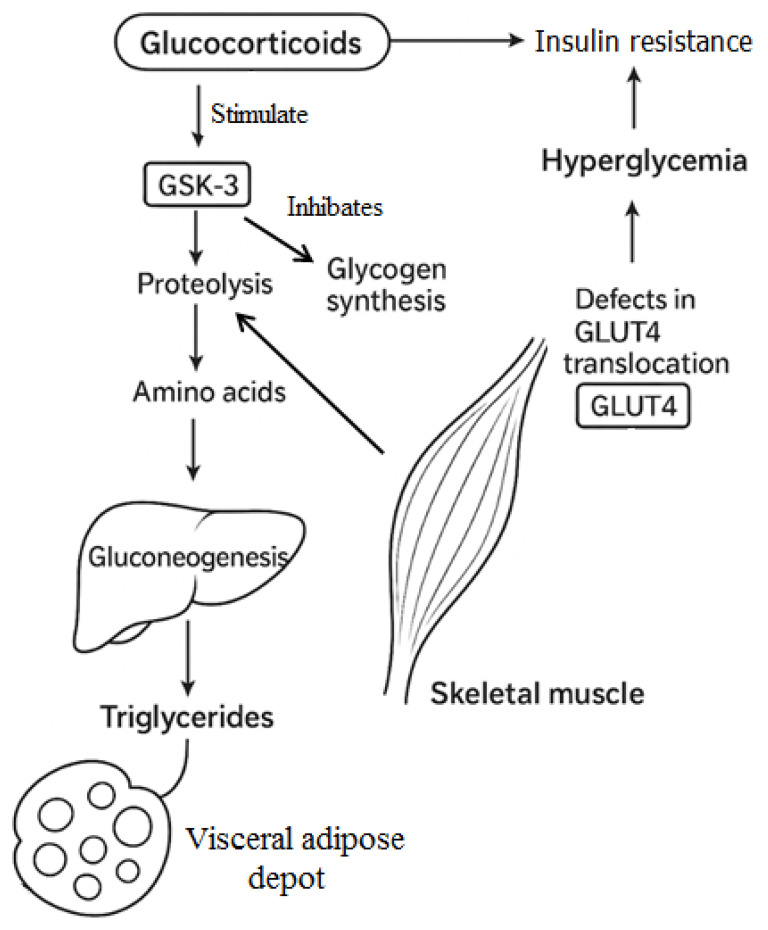
The effects of glucocorticoid hormones in adipose tissue and muscle.

**Figure 4 cimb-47-00565-f004:**
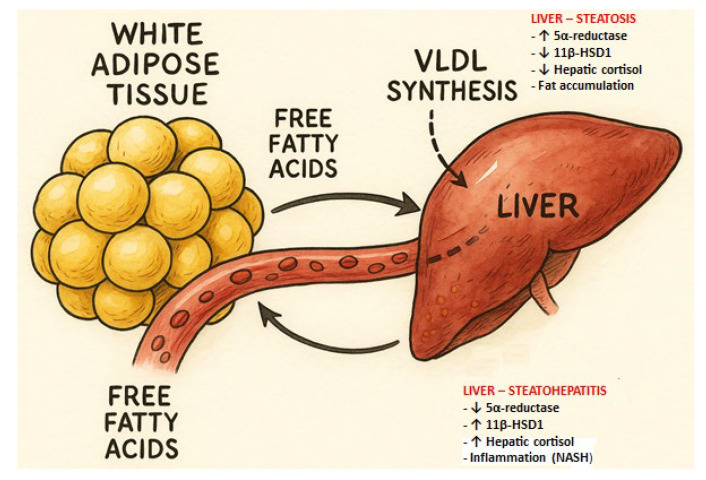
The liver–adipose tissue relationship. Hepatic glucocorticoid metabolism shifts during disease progression—from steatosis to steatohepatitis.

**Figure 5 cimb-47-00565-f005:**
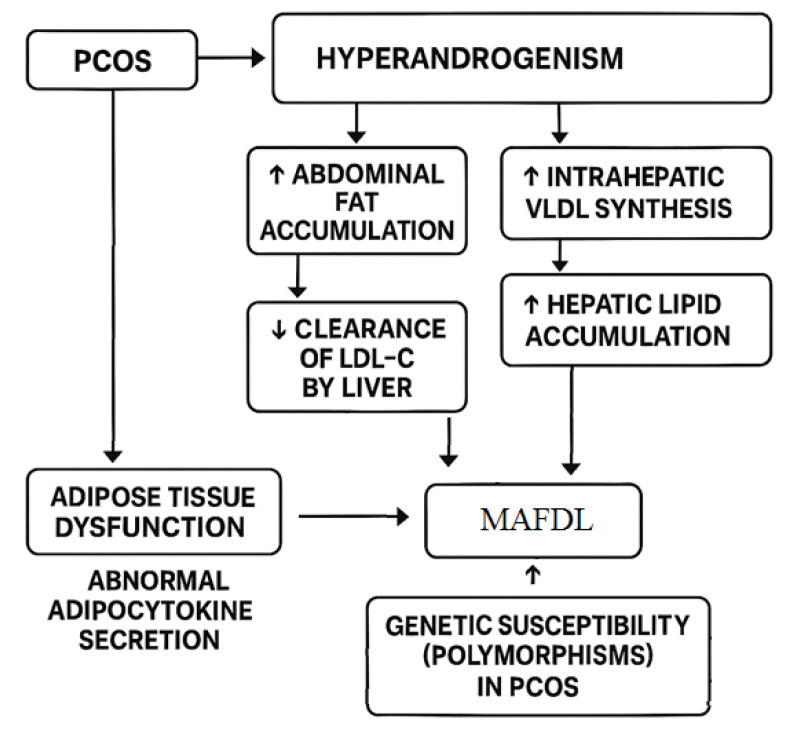
The link between PCOS and MAFLD.

**Figure 6 cimb-47-00565-f006:**
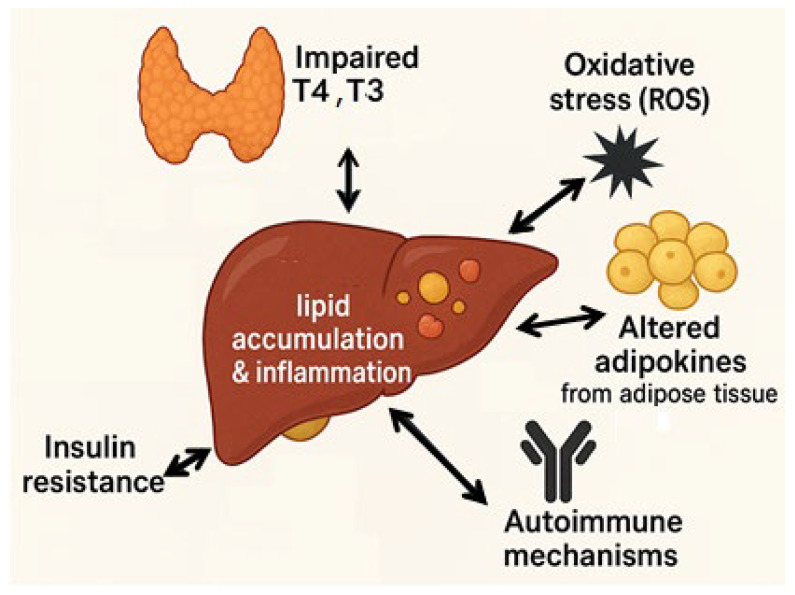
Bidirectional relationship between thyroid dysfunction and MASLD.

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
