# Peer review of "Lipid Hormones at the Intersection of Metabolic Imbalances and Endocrine Disorders"

_cimb, 2025, doi:10.3390/cimb47070565_

Round 1
Reviewer 1 Report
Comments and Suggestions for Authors
Current Issues in Molecular Biology
This is a well-written review and worth publishing.
Only minor editorial revisions are suggested.
Minor editorial comments are as follows:
- Line 20: LDL should be low density lipoprotein
- Lines 54, 73 and throughout the text: diabetes mellitus should be italic?
- Line 57: free fatty acids should be free fatty acids (FFA)
- Line 68: Glucocorticoids should be GCs
- Line 69: GCs play a critical…
- Lines 93, 101: (figure 1) (figure 2) should be (Figure 1) (Figure 2)
- Line 104: FFAs should be FFA ?
- Line 108: glucocorticoids (GCs) should be GCs
- Line 118: glucocorticoids should be GCs
- Line 119: glucocorticoid receptor should be GR
- Line 122: Glucocorticoids should be GCs
- Line 129 and throughout the text: de novo should be italic
- Line 132: glucocorticoid (GC) should be GC
- Lines 139-140: hypothalamic-pituitary-adrenal (HPA) should be HPA
- Line 151: glucocorticoid receptor (GR) should be GR
- Lines 167, 177: Glucocorticoids (GCs) should be GCs
- Line 179: (figure 3) should be (Figure 3)
- Lines 184, 199: free fatty acids (FFAs) should be FFA
- Lines 188, 201, 205: glucocorticoid should be GC
- Lines 236, 247: Glucocorticoids should be GCs
- Line 247: glucocorticoid should be GC
- Line 259: Glucocorticoids (GCs) should be GCs
- Line 260: glucocorticoid receptors should be GR
- Lines 274, 278: glucocorticoid should be GC
- Line 281: Jumar et al. [58] proposed…
- Line 285: delete [58]
- Lines 290-291: Melia azadirachta should be italic
- Lines 299, 304: glucocorticoid receptor should be GR
- Lines 301, 315, 319, 320: glucocorticoid should be GC
- Line 315: in vitro should be italic
- Line 331: non-alcoholic fatty liver disease (NAFLD) should be NAFLD
- Line 333: metabolic dysfunction-associated steatotic liver disease (MASLD) should be MASLD
- Line 335: diabetes mellitus should be italic ?
- Line 364: free fatty acids (FFAs) should be FFA
- Lines 396-397, 522: metabolic dysfunction associated steatotic liver disease (MASLD) should be MASLD
- Lines 428, 440: Thyroid Hormones should be THs
- Line 439: FT4 should be free T4 (FT4)
- Lines 441, 458:, 466, 470, 505, 538, 540 Thyroid hormone should be TH
- Lines 442, 467: free fatty acids should be FFA
- Lines 443, 444, 446
- : de novo should be italic?
- Lines 461, 500, 524: Thyroid hormones should be THs
- Lines 512, 515: diabetes mellitus should be italic ?
- Line 531: thyroid hormone receptor (THR) should be THR
- References: follow the format of Curr. Issues Mol. Biol.
For example: Hruby, A.; Hu, F.B. The epidemiology of obesity: A big picure. Pharmacoeconomics 2015, 33 (7), 673-689.
Author Response
For review article
Response to Reviewer 1 Comments
|
Dear Reviewer 1 and Editors,
Thank you very much for your positive assessment of our manuscript and for considering it worthy of publication in Current Issues in Molecular Biology.
We appreciate your careful reading and the detailed editorial suggestions provided. We have implemented all the recommended changes throughout the manuscript. These modifications are highlighted in blue for easy reference.
Additionally, we have ensured that the references now follow the journal’s required format.
We trust that the revised manuscript meets your expectations and is suitable for publication. Please do not hesitate to let us know if any further adjustments are needed.
Best regards,
Associate Professor Bogdana Virgolici, MD, PhD
Reviewer 2 Report
Comments and Suggestions for Authors
In the review the authors need to discuss about the role of inflammation and immune abnormalities and their modulation by various neuroendocrine hormones and their interactions.
Comments on the Quality of English Languageok
Author Response
For review article
Response to Reviewer 2 Comments
|
Dear Reviewer 2 and Editors,
Thank you very much for your time and for your thoughtful review of our manuscript entitled “Lipid Hormones at the Intersection of Metabolic Imbalances and Endocrine Disorders.”
We appreciate your suggestion to include a discussion on the role of inflammation and immune abnormalities, and their modulation by neuroendocrine hormones and interactions. We fully agree that these are important topics with significant clinical implications.
However, we respectfully believe that incorporating this additional layer of analysis would substantially broaden the scope of our current review. Our manuscript is specifically focused on the metabolic impact of lipid hormones—including glucocorticoids, thyroid hormones, and sex hormones—in relation to insulin resistance and MASLD. Our aim was to provide a concise and coherent narrative within this framework.
We have already addressed some aspects of inflammation where directly relevant (e.g., the paradoxical pro-inflammatory effects of chronic glucocorticoid excess, the role of adiponectin, and inflammatory signaling pathways such as NF-κB in hepatic steatosis). Nevertheless, a comprehensive exploration of neuroendocrine-immune interactions would merit a dedicated review article due to its complexity and scope.
Regarding your note on the English language, we have carefully revised the manuscript to improve clarity and expression throughout.
We hope that you understand our rationale for maintaining the current focus of the review and that the revised manuscript meets your expectations. We remain open to further suggestions you may have.
Best regards,
Associate Professor Bogdana Virgolici, MD, PhD
Reviewer 3 Report
Comments and Suggestions for Authors
Dobre et al. present a logical integration of the role of imbalances of glucocorticoid, thyroid, and sex hormones in metabolic dysfunction–associated steatotic liver disease (MASLD). The review is strong because it has a thematic progression logically—incrementally from complex molecular mechanisms of gluconeogenesis, lipolysis, and adipose dysfunction induced by glucocorticoids to sex hormone impact in PCOS and the beginning of thyroid hormone receptor–targeted therapies. Their inclusion of both basic mechanistic studies (e.g., 11β‑HSD1 transgenic mice) and clinical findings (e.g., early resmetirom trials) offers a rich context for drawing together endocrine dysregulation and hepatic steatosis, and the explanatory unity of a "shared metabolic basis" helps readers integrate disparate findings into a single scheme.
The review, however, would be improved by more balance in depth among hormone classes and a sharper critical sense of translational application. The glucocorticoid section, although thorough, overwhelms the relatively short thyroid and estrogen tests; in‑depth consideration of thyroid receptor isoform–selective actions and mechanisms of hepatic protection by estrogen would redress this imbalance. Therapeutic testing, for example, metformin in PCOS and resmetirom for MASLD, now reads lists of drugs without considering off‑target risk, adherence limitations, or prolonged safety profiles. Similarly, the change to MASLD from NAFLD/MAFLD is referred to but not put into context with evidence on prognostic stratification or how reclassification alters the implication of conventional cohorts. Lastly, overdependence on in vitro and rodent models is burdensome enough to require a special acknowledgment of species difference in hormone receptor expression and an opportunity for bridging studies, e.g., human organoid or ex vivo liver perfusion systems.
In brief, I recommend major revisions to tighten the manuscript's rigor and clinical relevance. First, rebalance the coverage by enriching the thyroid and estrogen topics with additional recent human cohort data and receptor‐isoform data. Second, complement therapeutic critique by examining trial design, exclusion criteria, and issues of real‐world adherence. Third, enrich MASLD nomenclature history by employing comparative outcome data and close translational gaps by proposing human‐based model systems. With these refinements, this review not only summarizes what is already known but also makes actionable suggestions for researchers and clinicians tackling hormone‐driven MASLD.
Round 2
Reviewer 2 Report
Comments and Suggestions for Authors I read the authors response. I still feel that the authors need to address the involvement of inflammation and immune response and their modulation by corticosteroid and they roof hormones, insulin and adipose tin in a short 1-2 paragraphs.Author Response
Please see the attachment.
Reviewer 3 Report
Comments and Suggestions for Authors
The authors' careful and extensive revisions have greatly augmented the manuscript's translational focus, clinical import, and thematic coherence. The argument on the function of estrogen is now firm and well-integrated; however, the article can be strengthened further by briefly mentioning the prospect of sex-specific identification of biomarkers, like employing metabolomics approaches (e.g., reference 91), in clinical conditions. This would once again reaffirm the translational importance of sex-differentiated results and further ground the mechanistic results in ultimate diagnostic methods. Secondly, although the mention of organ-on-chip and preclinical models based on artificial intelligence is appreciated, it would be beneficial to elaborate on this information. This can be transformed into a new subsection of future directions to continue the manuscript's forward-looking tone and stress the importance of human-relevant platforms to advance the field.
For final edit revision, there are two minor suggestions proposed before acceptance. First, Section 3.2 should have 1–2 sentences that explicitly emphasize the promise of sex-specific biomarkers. Second, Section 5's final paragraph should be rewritten more forcefully to advocate stronger use of organ-on-chip systems and AI as imperative tools in the validation of hormone-driven MASLD pathways. With these final adjustments, the review will finally meet the journal's standards in terms of originality, balance, and translational relevance.
Author Response
For review article
Response to Reviewer 3 Comments
|
Dear Reviewer 3 and Editors,
Thank you very much for your thorough review of our manuscript and for your valuable suggestions to enhance its quality and scientific depth.
In response to your comments:
- Rebalancing the coverage of estrogen and thyroid topics with additional recent human cohort data and receptor-isoform information
We have added a new section (3.2) on the involvement of estrogens and their mechanisms of action in MAFLD, including hormonal changes during premenopause and menopause and their connection to MAFLD. This section now incorporates updated references (83–91). Additionally, we introduced section 4.3, “MAFLD and Hypothyroidism: An Overlooked Endocrine Link,” highlighting the significance of both overt and subclinical hypothyroidism in the context of MAFLD. Furthermore, in section 4.4, “Therapeutic Strategies Targeting THR,” we elaborated on thyroid hormone receptor isoforms and emphasized the importance of understanding their isoform-specific actions for the development of targeted, organ-selective therapies for metabolic conditions such as MAFLD and NASH (new references 138–146). - Complementing therapeutic critique by examining trial design, exclusion criteria, and real-world adherence issues
In section 3.2, with references 93–95, we have described several studies on the role of metformin in PCOS, highlighting for each study the specific outcomes, advantages, and limitations. Similarly, in section 4.4, we have discussed the therapeutic effects and limitations of resmetirom and other thyromimetic agents in MAFLD, detailing aspects such as trial design, exclusion criteria, and challenges related to real-world adherence (references 147–153). - Enriching MASLD nomenclature history by employing comparative outcome data and proposing human-based model systems
At the beginning of the manuscript, supported by references 3–5, we clarified the distinction between MASLD and NAFLD and explained how MASLD refines the concept of MAFLD to support a more holistic and patient-centered approach. Moreover, we included relevant information on human-based model systems, such as the pancreas–liver microphysiological model, to bridge translational gaps in metabolic disease research.
We appreciate your insightful feedback, which has contributed significantly to improving the scope and clarity of our manuscript. We hope that the revisions meet your expectations and that the revised manuscript is now suitable for publication.
Best regards,
Associate Professor Bogdana Virgolici, MD, PhD